# The Association of Infection with Delirium in the Post-Operative Period after Elective CABG Surgery

**DOI:** 10.3390/jcm12144736

**Published:** 2023-07-17

**Authors:** Agnieszka Zukowska, Mariusz Kaczmarczyk, Mariusz Listewnik, Maciej Zukowski

**Affiliations:** 1Department of Infection Control, Regional Hospital Stargard, 73-110 Stargard, Poland; a.zukowska@op.pl; 2Sanprobi sp. z o.o. sp.k., 73-110 Szczecin, Poland; mariush@gmail.com; 3Department of Cardiac Surgery, Pomeranian Medical University, 70-111 Szczecin, Poland; sindbaad@poczta.onet.pl; 4Department of Anesthesiology, Intensive Care and Acute Intoxication, Pomeranian Medical University, 70-111 Szczecin, Poland

**Keywords:** infections, sepsis, CABG, delirium, long-term survival, cardiac surgery

## Abstract

Delirium is one of the most common complications of coronary artery by-pass graft (CABG) surgery. The identification of patients at increased risk of delirium and the implementation of preventive measures to reduce the risk of postoperative delirium is necessary to improve treatment outcomes after CABG. The aim of this study was to assess the association between postoperative delirium and postoperative infection and 10-year mortality in patients undergoing CABG surgery. This is a retrospective, observational cohort study of patients undergoing planned on-pump CABG between April 2010 and December 2012. We analysed a group of 3098 patients operated on in our cardiac surgery centre, from whom we selected a cohort of patients undergoing planned CABG surgery. All patients were assessed for postoperative infection, such as pneumonia, bloodstream infections (BSIs) and surgical site infections (SSIs). Patients who experienced postoperative delirium were significantly more likely to have infection (7.4% vs. 22%; *p* = 0.0037). As regards particular types of infection, significant differences were only found for pneumonia and sternal SSIs. Patients who experienced postoperative delirium had significantly lower 5-year (*p* = 0.0136) and 10-year (*p* = 0.0134) survival. Postoperative delirium significantly increases long-term mortality in patients undergoing CABG surgery. Pneumonia and sternal SSIs significantly increase the risk of postoperative delirium in cardiac surgery patients.

## 1. Introduction

According to data from the World Health Organization (WHO), 126 million people, approximately 1.72% of the world’s population, were affected by ischaemic heart disease (IHD) in 2017 [1]. In 2016, IHD was responsible for approximately 2.2 million deaths in Europe and more than 9 million deaths globally [2]. Thus, IHD remains one of the major challenges of modern medicine. Interventional procedures, including percutaneous coronary angioplasty with or without endovascular stent placement, and surgical myocardial revascularisation are important treatment options for patients with IHD. To date, coronary artery bypass graft (CABG) surgery performed with cardiopulmonary bypass (CPB) via median sternotomy has been the most common type of cardiac surgery. Cardiac surgery carries an increased risk of complications, which is due to its invasive nature and the activation of inflammatory response. Delirium is one of the most common complications of CABG surgery. According to the available literature, the incidence of delirium in patients undergoing CABG surgery ranges between 8% and 54% [3,4]. While postoperative delirium is usually reversible, it is associated with poorer prognosis, increased length of hospital stay and increased mortality [5]. The identification of patients at increased risk of delirium and the implementation of preventive measures to reduce the risk of postoperative delirium is necessary to improve treatment outcomes after CABG surgery. Several pharmacological and non-pharmacological strategies have been studied to prevent and treat postoperative delirium, with mixed results. Most studies were focused on potential-use preventive agents or avoidance delirium-induced drugs. A specific agent or medication class such as psychoactive agents can induce delirium and should be avoided in the preoperative period, especially opioids and benzodiazepines. Varied results were reported by studies that investigated the association between preoperative cardiovascular drugs, such as statins, b-blockers and others, use and postoperative delirium [6,7]. Melatonin and ramelteon have been studied as a preventive measure against the development of delirium in hospitalised patients with different results. However, recent systematic review and meta-analysis of randomised controlled trials shown that melatonin and ramelteon do not seem to reduce delirium incidence in ICU patients [8]. Nonpharmacological strategies of postoperative delirium prevention include early physical rehabilitation, occupational therapy, sleep improvement strategies, sensory deficit correction, environmental management and hydration. Many recent studies have shown significant benefits of nonpharmacological interventions in decreasing postoperative delirium [9,10]. The pathogenesis of postoperative delirium is most likely complex, i.e., a number of factors contribute to its development. The pathological mechanisms underlying postoperative delirium include inflammatory changes in the CNS, resulting from the activation of inflammatory responses caused by the surgical procedure itself or infection [11,12]. Moreover, a number of studies have shown that infection is associated with an increased risk and greater severity of delirium. However, most of these studies concerned urinary tract infections, and the association between postoperative delirium and postoperative infection after CABG surgery has yet to be clarified. The aim of this study was to assess the association between postoperative delirium and postoperative infection and 10-year mortality in patients undergoing CABG surgery.

## 2. Materials and Methods

This is a retrospective, observational cohort study of patients undergoing planned on-pump CABG in the Department of Cardiac Surgery of the Pomeranian Medical University between April 2010 and December 2012. Written informed consent was not required, as the study was retrospective in nature. The study protocol was in line with the ethical guidelines of the Declaration of Helsinki of 1964. We analysed a group of 3098 patients operated in our cardiac surgery centre, from whom we selected a cohort of patients undergoing planned CABG surgery. We excluded patients with pre-existing cognitive dysfunction, psychiatric disorders and neurological conditions (myasthenia gravis, Parkinson’s disease, etc.). The patients were followed-up to identify delirium for 30 days after surgery. All patients were assessed for postoperative infection, such as pneumonia, bloodstream infections (BSIs) and surgical site infections (SSIs), in accordance with ECDC (European Centre of Disease Control) criteria. In addition, preoperative, perioperative and postoperative clinical data were collected. Preoperative data included age, sex, weight, height, smoking status, comorbidities, left ventricular ejection fraction (LVEF), NYHA functional class, EuroSCORE, logistic EuroSCORE and preoperative laboratory parameters such as haemoglobin (Hb) concentration, haematocrit (Hct) value, platelet (PLT) count, HbA1c value and serum creatinine level. The following perioperative parameters were analysed: duration of CPB, duration of surgery, use of pressor amines, need for an intra-aortic balloon pump (IABP), intraoperative ultrafiltration, postoperative LVEF, fluid balance (days 1 and 2 after surgery), volume of postoperative drainage, need for RBC and PLT transfusion as well as laboratory parameters such as haemoglobin concentration, haematocrit value, platelet count and serum creatinine level within 1 day after surgery and CK-MB value at 6, 12 and 24 h after surgery. The postoperative course was analysed with respect to postoperative complications, such as TIA, MI, stroke, AF and infection, within 30 days after surgery. We initially compared demographic and preoperative data between patients with and without postoperative delirium. We then analysed the association between perioperative parameters and the incidence of postoperative delirium. In the next step of our analysis, we compared the incidence of postoperative complications between patients who experienced postoperative delirium and those who did not experience postoperative delirium. In the final step of our analysis, we assessed the impact of postoperative infection on postoperative delirium. In addition, we obtained data on the 5-year and 10-year survival of the patients included in our study from the Polish National Cardiac Surgery Registry and analysed the impact of postoperative delirium on 5-year and 10-year survival after CABG surgery. The primary outcome was the incidence of post operative delirium and secondary long-term survival.

Survival curves were created using the Kaplan–Meier method. The Cox regression model was used to quantify the effect size. It was used to fit both univariable and multivariable regression models. The effect size was expressed as a hazard ratio (HR) with 95% confidence intervals. All variables that were significant in univariable analysis were included in the multivariable analysis. Survival analysis was conducted using the survival package in R (https://cran.r-project.org, accessed on 28 March 2023). Statistical significance was set at *p* < 0.05.

## 3. Results

Of the 3098 patients analysed, 778 patients undergoing planned CABG surgery were ultimately included in the study (Figure 1). Twenty-four per cent (*n* = 187) of the patients were women. 

The mean age of the patients was 65 years. A total of 240 patients (30.8%) were active smokers at the time of admission to hospital. A comparison of all demographic and preoperative data between patients who experienced delirium and those who did not experience delirium is shown in Table 1.

Statistically significant differences were only found for the following parameters: age, ES and ESlog. Patients who experienced postoperative delirium were significantly older (64 vs. 70 years) and had a higher EuroSCORE and a higher logistic EuroSCORE, which indicates that their preoperative condition was worse compared to patients who did not experience postoperative delirium. Data on comorbidities in patients with delirium and those without delirium are shown in Table 2.

No significant differences in terms of comorbidities were found between patients who experienced delirium and those who did not experience delirium.

The mean duration of surgery for all patients was 151.8 min, whereas mean duration of CPB was 50.2 min. Four patients required intra-aortic counterpulsation, and twenty patients received milrinone infusion. All perioperative data are shown in Table 3. 

No significant differences were found between patients with and without delirium in terms of perioperative parameters. One exception was mean intraoperative ultrafiltration volume, which was significantly greater in patients who experienced delirium compared to patients who did not experience delirium (351 mL vs. 148 mL; *p* = 0.0215). A comparison of data on the postoperative course between patients with delirium and those without delirium is shown in Table 4.

Patients who experienced postoperative delirium were significantly more likely to have infection (7.4% vs. 22%; *p* = 0.0037). As regards particular types of infection, significant differences were only found for pneumonia and sternal SSIs.

Postoperative delirium was associated with poorer long-term prognosis. Patients who experienced postoperative delirium had significantly lower 5-year (*p* = 0.0136) and 10-year (*p* = 0.0134) survival; Kaplan–Meier curves are shown in Figure 2 and Figure 3.

## 4. Discussion

Despite numerous studies, the pathological mechanisms underlying postoperative delirium have not been fully elucidated. However, researchers agree that patients undergoing cardiac surgery are at high risk of this complication. The incidence of delirium in the patients included in the present study was relatively low (7.6%) compared to the incidence rates reported by other authors. A study by Deninger et al. based on the data of 1538 cardiac surgery patients showed that postoperative delirium developed in 22.3% of the patients, whereas a meta-analysis by Lin et al. showed that delirium was diagnosed in 17% of the 19,785 patients included in the studies analysed [13,14]. The difference may be due to the fact that our study only included patients who were scheduled for planned CABG surgery, whose comorbidities were well controlled before surgery and who were in a relatively good overall condition. A number of independent risk factors for postoperative delirium have been identified, the most important being age, diabetes, TIA, electrolyte disturbances and renal failure [13,14,15,16]. In a 2021 meta-analysis by Chen et al. analysing 14 studies with a total sample size of 13,286, the following risk factors for postoperative delirium in cardiac surgery patients were identified: age, carotid artery stenosis, depression, diabetes, NYHA functional class III or IV, duration of mechanical ventilation and length of ICU stay [17]. In their 2022 meta-analysis on postoperative delirium after transcatheter aortic valve replacement, Ma et al. identified the following risk factors for delirium: increased age, male sex, prior stroke or TIA, atrial fibrillation/flutter, weight loss, electrolyte abnormality, general anaesthesia and postoperative acute kidney injury [18]. Similarly, in our study, older age and worse overall preoperative condition were found to be associated with a higher incidence of postoperative delirium. While the incidence rate of delirium in the patients included in the present study was relatively low (7.6%), our finding was statistically significant, which is consistent with other publications. While the present study did not find an association between NYHA functional class and postoperative delirium, it demonstrated a significant relationship between postoperative delirium and EuroSCORE and logistic EuroSCORE, which is consistent with observations that preoperative cardiovascular health has a significant impact on the postoperative course, including postoperative delirium. In 2022, Andrási et al. published the results of their study, which showed that an increased need for RBC and fresh frozen plasma transfusion as well as a longer duration of surgery and a longer duration of CPB were associated with an increased risk of postoperative delirium [19]. In contrast, a 2023 analysis of the DECADE trial carried out by Sari et al., which included 585 Cleveland Clinic patients undergoing cardiac surgery, did not find an association between postoperative haemoglobin level and the incidence of postoperative delirium [16]. Our study did not find an association between the intraoperative course and postoperative delirium, which may be due to the cohort selected and a small number of intraoperative complications leading to increased duration of surgery. A number of publications have indicated that there is an association between infection and postoperative delirium. However, there are no data to confirm this association in patients undergoing cardiac surgery. In their study published in 2017, Kuswardhani et al. noted that in elderly individuals, delirium may be triggered by a number of factors, including infection such as UTIs, pneumonia and sepsis [20]. In their 2022 systematic review, Dutta et al. noted that infection (particularly UTIs and pneumonia) is the most common triggering factor of delirium [21]. UTIs are the most researched type of infection, and their association with delirium has been most extensively investigated [22]. In our study, infection was found to be a factor predisposing to postoperative delirium. However, our analysis of particular types of infection showed that pneumonia and sternal SSIs were the only types of infection that were associated with an increased incidence of postoperative delirium. Not surprisingly, a generalised inflammatory response can cause a whole range of central nervous system disturbances associated with a surge of cytokines and disturbances in optimal oxygenation of the brain [23]. It should be stressed that both pneumonia and sternal SSIs may result in impaired ventilation and thus in impaired gas exchange, followed by hypoxia, leading to CNS disturbances, including delirium. The role of hypoxia in the development of delirium is well established. Spiropoulou et al. found in their study that hypoxemia after extubation significantly increases the risk of postoperative delirium (OR = 20.6; 95% CI: 2.82–150) [24]. Both delirium and infection significantly worsen prognosis in patients with any underlying disease. For instance, Pannone et al. found in their study that patients diagnosed with CDI (Clostridium difficile infection) who developed delirium were at the highest risk for dying within 30 days of CDI diagnosis [25]. Postoperative delirium leads to negative consequences, both in terms of cognitive impairment and quality of life, in patients undergoing cardiac surgery [26]. A meta-analysis by Lin et al. showed that postoperative delirium is associated with increased short-term and long-term mortality [14]. Our findings are in line with these observations. Our Kaplan–Meier analysis showed that those patients included in the present study who experienced delirium had significantly lower 5-year and 10-year survival. Delirium inevitably leads to somatic symptoms, which is due to worse self-care and impaired postoperative rehabilitation, with all the consequences of eating disorders and many others. Exposure to antipsychotics often results in side-effects, such as increased pneumonia risk [27]. Thus, it is believed that standardised pharmacological management of postoperative delirium may improve patients’ treatment outcomes and quality of life [12]. Some studies reported that capillary refill is significantly associated with postoperative delirium; however, this relationship remains unclear. However, it’s important to remember that preoperative assessment of capillary refill time would be an easy method to investigate peripheral perfusion before surgery, which could have influenced the incidence of postoperative delirium [28,29,30,31]. This study also has several limitations. First, this is a single hospital study. The use of administrative hospital databases introduced an inherent bias that should be taken into consideration. We also do not control for post-hospitalisation confounders that might have a causal relationship with long-term mortality, as this is limited by our data sources.

## 5. Conclusions

In this study, postoperative delirium significantly increases long-term mortality in patients undergoing CABG surgery. Pneumonia and sternal SSIs significantly increase the risk of postoperative delirium in cardiac surgery patients.

## Figures and Tables

**Figure 1 jcm-12-04736-f001:**
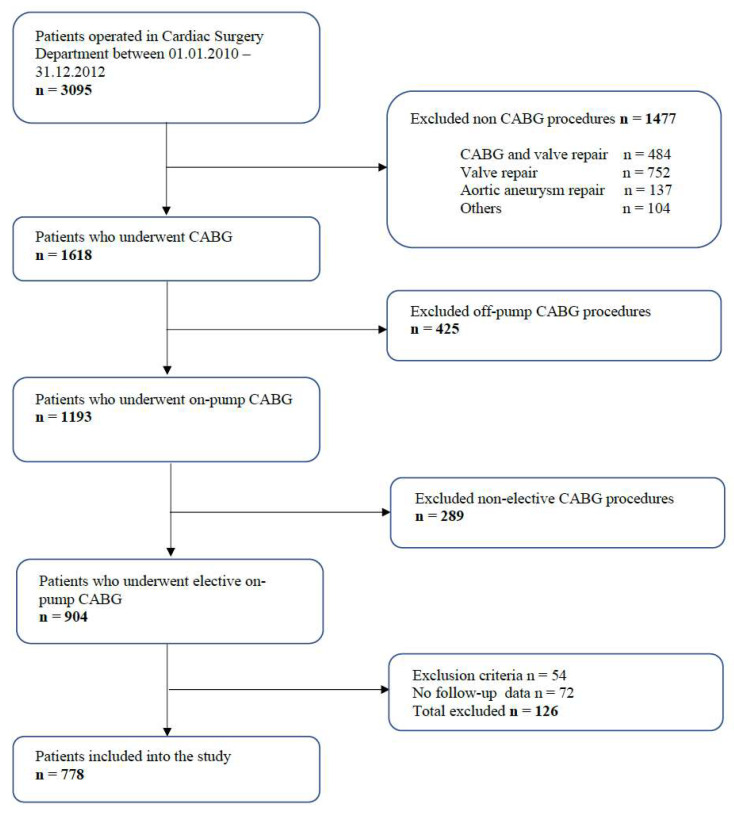
Patient flow chart.

**Figure 2 jcm-12-04736-f002:**
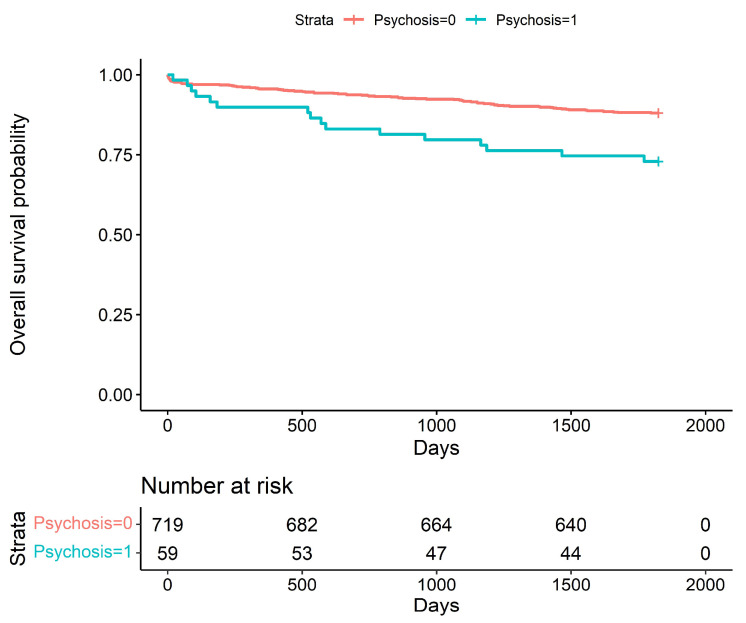
Kaplan–Meier curve, comparison of two groups (delirium vs. no delirium)—5-year follow-up (*p* = 0.0136).

**Figure 3 jcm-12-04736-f003:**
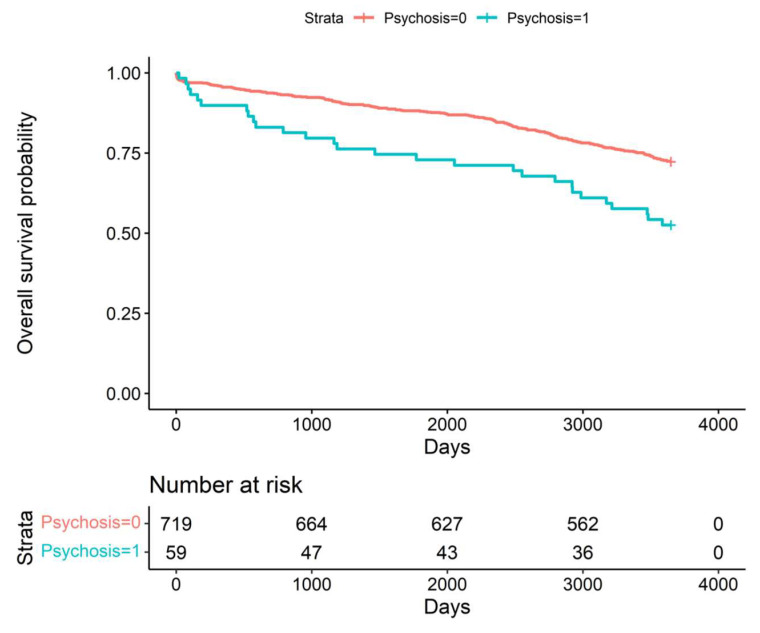
Kaplan–Meier curve, comparison of two groups (delirium vs. no delirium)—10-year follow-up (*p* = 0.0134).

**Table 1 jcm-12-04736-t001:** Demographic data and preoperative condition.

Parameter	Delirium (−) *n* = 719	Delirium (+) *n* = 59	*p*
Female (*n*. %)	170/719 (24%)	17/59 (29%)	0.9999
Age (years. mean ± SD)	64 (9)	70 (10)	0.0002
Weight (kg. mean ± SD)	81 (13)	78 (11)	0.3110
BMI	29.3 (8.4)	28.8 (3.6)	0.6530
Smoking (n. %)	312/719 (11)	14 (59)	0.0602
ES (mean ± SD)	3.77 (2.69)	5.32 (2.41)	<0.0001
ESlog (mean ± SD)	4.7 (5.3)	6.7 (4.7)	<0.0001
LVEF (%. mean ± SD)	50 (11)	48 (11)	0.5801
NYHA functional class (n. %)			0.9999
1	3/706 (0.4%)	1/59 (1.7%)	
2	138/706 (20%)	10/59 (17%)	
3	400/706 (57%)	36/59 (61%)	
4	165/706 (23%)	12/59 (20%)	
Insulin intake (*n*. %)	101/718 (14%)	10/59 (17%)	0.7277
HbA1c (%, mean ± SD)	2.11 (3.25)	3.18 (3.80)	0.1557
Hb (mg/dL, mean ± SD)	8.45 (0.79)	8.17 (0.76)	0.0711
Hct (%, mean ± SD)	0.41 (0.04)	0.39 (0.04)	0.1577
PLT (mean ± SD)	246 (66)	261 (72)	0.4609
Serum creatinine level mg/dL (mean ± SD)	0.95 (0.50)	1.07 (1.08)	0.5478

BMI—body mass index, ES—EuroSCORE, ESlog—logistic EuroSCORE, LVEF—left ventricular ejection fraction, NYHA—New York Heart Association, HbA1c—glycated haemoglobin, Hb—haemoglobin, Hct—haematocrit, PLT—platelets.

**Table 2 jcm-12-04736-t002:** Comorbidities.

Parameter	Delirium (−) *n* = 719	Delirium (+) *n* = 59	*p*
Insulin dependent diabetes (n. %)	8/719 (1.1%)	0/59 (0%)	0.9999
Insulin non-dependent diabetes (n. %)	225/719 (31%)	26/59 (44%)	0.3664
Prior stroke (n. %)	32/719 (4.5%)	5/59 (8.5%)	0.9999
TIA (n. %)	6/719 (0.8%)	0/59 (0%)	0.9999
Hypertension (n. %)	552/719 (77%)	52/59 (88%)	0.2622
Myocardial infarction (n. %)	245/718 (34%)	19/59 (32%)	0.9999
Atrial fibrillation (n. %)	23/719 (3.2%)	3/59 (5.1%)	0.9999
Hypercholesterolemia (n. %)	57/719 (7.9%)	3/59 (5.1%)	0.9999
Dyslipidemia (n. %)	66/719 (9.2%)	2/59 (3.4%)	0.8493
Peripheral arterial disease (n. %)	77/719 (11%)	10/59 (17%)	0.8493
Renal failure (n. %)	31/719 (4.3%)	3/59 (5.1%)	0.9999
Autoimmune diseases (n. %)	14/719 (1.9%)	1/59 (1.7%)	0.9999

TIA—transient ischaemic attack.

**Table 3 jcm-12-04736-t003:** Perioperative course.

Parameter	Delirium (−) *n* = 719	Delirium (+) *n* = 59	*p*
Duration of surgery (min. mean ± SD)	152 (50)	150 (22)	0.9973
Duration of CPB (min. mean ± SD)	50 (15)	49 (13)	0.9973
CPB fluid balance (mL. mean ± SD)	572 (613)	498 (749)	0.7277
IntraoperativeUF (mL. mean ± SD)	148 (469)	351 (671)	0.0215
LVEF day after surgery %	52 (10)	49 (11)	0.1503
Day 1 fluid balance (mL. mean ± SD)	−1105 (1060)	−807 (1109)	0.2682
Day 2 fluid balance (mL. mean ± SD)	−179 (849)	−115 (874)	0.5801
Overall fluid balance (mL. mean ± SD)	−1255 (1429)	−928 (1291)	0.1557
Intra-aortic balloon pump (n. %)	4/716 (0.6%)	0/59 (0%)	0.9999
Milrinone (n. %)	17/704 (2.4%)	3/58 (5.2%)	0.9999
Postoperative drainage (mL. mean ± SD)	545 (335)	559 (397)	0.9973
Postoperative Hb (mg/dL, mean ± SD)	6.72 (0.81)	6.63 (0.75)	0.5801
Postoperative Hct (%, mean ± SD)	0.34 (0.05)	0.32 (0.04)	0.4609
Postoperative PLT (,mean ± SD)	189 (54)	198 (51)	0.4609
Postoperative Cr level mg/dL (mean ± SD)	1.03 (0.66)	1.14 (0.61)	0.0711
RBC (mL. mean ± SD)	138 (288)	174 (296)	0.5478
PLT (mL. n. %)			0.9973
0	599/717 (84%)	48/59 (81%)	
200	1/717 (0.1%)	0/59 (0%)	
250	60/717 (8.4%)	4/59 (6.8%)	
300	47/717 (6.6%)	7/59 (12%)	
500	2/717 (0.3%)	0/59 (0%)	
550	5/717 (0.7%)	0/59 (0%)	
600	3/717 (0.4%)	0/59 (0%)	
CK-MB after 6 h	38 (25)	42 (32)	0.6933
CK-MB after 12 h	44 (46)	54 (49)	0.4609
CK-MB after 24 h	48 (63)	54 (47)	0.4675

CPB—cardiopulmonary bypass, LVEF—left ventricular ejection fraction, UF—ultrafiltration during CPB, Hb—haemoglobin, Hct—haematocrit, PLT—platelets, Cr—serum creatinine, FFP—fresh frozen plasma, CK-MB—creatine kinase-myocardial band.

**Table 4 jcm-12-04736-t004:** Postoperative course.

Parameter	Delirium (−) *n* = 719	Delirium (+) *n* = 59	*p*
Postoperative TIA (n. %)	4/719 (0.6%)	1/59 (1.7%)	0.9999
Postoperative MI (n. %)	12/719 (1.7%)	1/59 (1.7%)	0.9999
Postoperative stroke (n. %)	6/719 (0.8%)	1/59 (1.7%)	0.9999
Postoperative AF (n. %)	161/719 (22%)	16/59 (27%)	0.9999
Infection (n. %)	53/719 (7.4%)	13/58 (22%)	0.0037
Infection source (n. %)			0.0025
0	666/719 (93%)	45/58 (78%)	
BSI	14/719 (1.9%)	1/58 (1.7%)	
SSI sternal	10/719 (1.4%)	5/58 (8.6%)	
SSI leg	2/719 (0.3%)	1/58 (1.7%)	
Pneumonia	27/719 (3.8%)	6/58 (10%)	

TIA—transient ischaemic attack, MI—myocardial infarction, AF—atrial fibrillation, BSI—blood stream infection, SSI—surgical site infection.

## Data Availability

All data will be available for reasonable request.

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
