# Peer review of "The Association of Infection with Delirium in the Post-Operative Period after Elective CABG Surgery"

_jcm, 2023, doi:10.3390/jcm12144736_

Round 1

Reviewer 1 Report

The publication addresses in a very interesting way the topic of links between the occurrence of healthcare-associated infection and postoperative delirium in patients undergoing cardiac surgery. The authors analyzed not only the incidence of infections, but also 5 and 10-year mortality, which turned out to be higher among patients with delirium. Deep SSI or pneumonia has been shown to increase the risk of postoperative delirium. The results of the presented research may lead to an increase of the standard of care for patients undergoing cardiac surgery by increasing the supervision of patients with HAI symptoms for the occurrence of delirium symptoms. In my opinion just minor language corrections are needed - line 64 "operated on in our" - "on" is not needed and they should decide which version is used in the manuscript "dyslipidemia" (~aemia) or "dyslipidemia" "hypercholesterolaemia" or "hypercholesterolemia" (~emia).

Very good with minor changes listed in the previous section.

Author Response

 Responses to Reviewers
July 12, 2023
Dear Editor and Reviewers,
Re: jcm-2477685

We are grateful to the Editorial Board for considering a revised version of our manuscript entitled “The association of infection with delirium in post-operative period after elective CABG surgery.”.We appreciate the time and effort that you and the reviewers dedicated to providing feedback on our manuscript and we are grateful for the insightful comments on and valuable improvements to our paper. We have taken into account the careful and thoughtful comments of the Editors and Reviewers and revised the manuscript in line with their recommendations.     
Please find below a detailed point-by-point reply to the comments made by editors and reviewers:

  • The pharmacological and non-pharmacological strategies to prevent and treat postoperative delirium were discussed in introduction section, and all suggested references were added.
  • Outcomes were clarified in the methods section.
  • Thank you very much for your careful reading and for your attention, figure 1 was moved it to the appropriate section.
  • The preoperative assessment of capillary refill time was the element of the standard preoperative examination; however, it was not documented, that why we didn’t include it in our study. Despite the capillary refill time could have influenced the incidence of postoperative delirium this relationship remains unclear, that why we don’t agree that is the limitation of the study. However, there is very interesting subject, we discussed it in discussion section, and cited all suggested references.
  • The statistical analysis details were added in the methods section, is wasn’t in previous version only due to editorial error, thank you once again for careful reading and very helpful suggestion.
  • The suggested sentence "In this study," before "Postoperative delirium significantly increases long-term mortality in patients undergoing CABG surgery" was added in the conclusions.
  • This study also has several limitations. First, this is a single hospital study. The use of administrative hospital databases introduced an inherent bias that should be taken into consideration. Capillary refill time was not recorded, and we were thus unable to add it to statistical analysis. We also do not control for post hospitalization confounders that might have a causal relationship with long term mortality, as this is limited by our data sources. The limitation part was added at the end of discussion section.
  • All suggested language corrections were done, we decided to use "dyslipidemia" and “hypercholesterolemia” version, thank you very much for the careful and thoughtful comments.

 The changes are marked up using the “Comments” function, and the revised manuscript was resubmitted.       
We hope our research manuscript will now be suitable for publication on Journal of Clinical Medicine. We would like to thank the referee again for taking the time to review our manuscript.

Sincerely yours,

Maciej Zukowski, Prof MD, PhD

Reviewer 2 Report

- Line 46-48. Authors should add that, to date, several pharmacological (doi: 10.3390/jcm12020435 - doi: 10.1186/s12877-017-0695-x - doi: 10.1186/s40560-017-0224-1) and non-pharmacological strategies (doi: 10.3389/fmed.2023.1099594 - doi: 10.7759/cureus.10096) have been studied to prevent and treat postoperative delirium, with mixed results. Please briefly discuss and add these 5 references.

- Please clarify the primary and secondary investigated outcomes in the methods section.

- Line 67. Figure 1 should be part of the results section. Please move it to the appropriate section.

- Did authors perform preoperative assessment of capillary refill time?

- Please include the details of the statistical analysis used in the methods section.

- Please write a sub-paragraph of the discussion with the limitations of the study.

- In this paper, assessment of Capillary Refill Time would have been an easy tool (doi: 10.1186/s12871-022-01920-1) to investigate peripheral perfusion before surgery, which could have influenced the incidence of postoperative delirium (doi: 10.1136/emj.2007.055244 - doi: 10.1186/s12871-020-01171-y). Please discuss it as a limitation and add these 3 references.

- Please add "In this study," before "Postoperative delirium significantly increases long-term mortality in patients undergoing CABG surgery" in the conclusions.

Author Response

(The authors gave the same response as above.)

Round 2

Reviewer 2 Report

Dear authors, after the revision I have no more comments to make.